Perception of global facial geometry is modulated through experience

Ramon Meike meike.ramon@uclouvain.be Meike.Ramon@glasgow.ac.uk
Institute of Research in Psychology, Institute of Neuroscience, Université catholique de Louvain , Louvain-La-Neuve , Belgium
Institute of Neuroscience and Psychology, University of Glasgow , Glasgow , United Kingdom
Tavano Alessandro
Electronic publication date: 2015 Mar 24
Publication date: 2015
Volume: 3
Electronic Location ID: e850
Received 2014 Nov 26; Accepted 2015 Mar 3
Copyright: © 2015 Ramon
Copyright year: 2015
Copyright holder: Ramon
License: This is an open access article distributed under the terms of the Creative Commons Attribution License, which permits unrestricted use, distribution, reproduction and adaptation in any medium and for any purpose provided that it is properly attributed. For attribution, the original author(s), title, publication source (PeerJ) and either DOI or URL of the article must be cited.
License URL: https://creativecommons.org/licenses/by/4.0/

Keywords: Personal familiarity, Face processing, Real-life exposure, Viewing distances, Holistic processing, Face geometry effect, Facial configuration

Funding: Belgian National Foundation for Scientific Research (FNRS) Meike Ramon is supported by the Belgian National Foundation for Scientific Research (FNRS). The funders had no role in study design, data collection and analysis, decision to publish, or preparation of the manuscript.

==============================
Identification of personally familiar faces is highly efficient across various viewing conditions. While the presence of robust facial representations stored in memory is considered to aid this process, the mechanisms underlying invariant identification remain unclear. Two experiments tested the hypothesis that facial representations stored in memory are associated with differential perceptual processing of the overall facial geometry. Subjects who were personally familiar or unfamiliar with the identities presented discriminated between stimuli whose overall facial geometry had been manipulated to maintain or alter the original facial configuration (see Barton, Zhao & Keenan, 2003). The results demonstrate that familiarity gives rise to more efficient processing of global facial geometry, and are interpreted in terms of increased holistic processing of facial information that is maintained across viewing distances.

Holistic face processing refers to the ability to simultaneously integrate facial information into a unified percept (Sergent, 1984; Young, Hellawell & Hay, 1987; McKone, Martini & Nakayama, 2003; Rossion, 2008). This ability, which other authors have coined configural or face interactive processing (Goffaux, 2009; for a review see e.g., Maurer, Le Grand & Mondloch, 2002), is considered a hallmark of human face processing expertise (Mondloch et al., 2007). Paradigms commonly used to measure holistic processing include the composite face illusion (Young, Hellawell & Hay, 1987), the whole-part advantage (Tanaka & Farah, 1993), or the face inversion effect (Yin, 1969), all of which demonstrate the inter-dependency of perceiving information in upright faces (for recent reviews see e.g., Rossion, 2009; Rossion, 2013).

Another paradigm developed in the face processing literature that relies on holistic face processing was reported by Barton, Zhao & Keenan (2003). They used faces of two unfamiliar identities that were modified to incorporate different types of combinations of positional shifts of facial features. In their oddity paradigm, each trial involved simultaneous presentation of three face stimuli depicting one of these two unfamiliar identities. Two of the three stimuli were identical, and subjects were required to identify the one that differed from them. The subjects’ discrimination performance was superior for combinations involving more severe distortions of the triangular relation of the mouth and eyes (e.g., eyes closer and mouth down), as compared to those that preserved the original aspect ratio (e.g., eyes farther apart and mouth down; see Fig. 1 for an illustration). In healthy observers, this face geometry effect was found for upright, but not inverted face discrimination. Moreover, the effect was not observed in the prosopagnosic patient TS, regardless of stimulus orientation. These findings were taken to reflect observers’ ability to “integrate local spatial information into overall facial structure.”

Figure 1 Examples of stimuli used in Experiments 1 and 2.

Stimulus manipulations included combined changes of the inter-ocular and nose-mouth distance, giving rise to the four possible combinations depicted here. Two types of changes were considered as “less” (black borders), or “more” distorting (gray borders), as the eyes and mouth were moved either in the same (eyes/mouth out, eyes/mouth in) or opposite (eyes out/mouth in, eyes in/mouth out) direction, thereby respectively preserving, or altering the original facial configuration.

We have previously reported similar findings in an investigation of personally familiar face processing in pure prosopagnosia (Ramon & Rossion, 2007; M Ramon, T Busigny, G Gosselin and B Rossion, unpublished data). Specifically, we tested PS, an extensively studied pure case of prosopagnosia (first reported by Rossion et al., 2003; see Rossion, 2014 for a recent review), and her colleagues who worked together as kindergarten teachers. Adding to Barton et al.’s (2003) findings, we found that healthy controls showed a face geometry effect during veridicality decisions between simultaneously presented original and altered faces of the ∼30 children they supervised. On the other hand, PS did not show this advantage for discrimination of more over less distorting changes of the overall facial configuration.

These independent observations are to our knowledge the only studies having used this paradigm. As both reported a lack of a face geometry effect in two cases of acquired prosopagnosia, we reasoned that the manipulations used to measure the face geometry effect may represent an alternative measure of holistic processing and face processing efficiency. However, subtle differences between the aforementioned studies give rise to a number of questions.

First, are healthy observers with normal holistic face processing abilities equally capable of discerning manipulations of facial geometry? The previous studies—using both unfamiliar and personally familiar faces—indicate that this might indeed be the case. However, the reported observations were made using two initially unfamiliar identities (Barton, Zhao & Keenan, 2003) presented repeatedly over a large number of trials, or a larger group of personally familiar ones encountered repeatedly in real-life situations (Ramon & Rossion, 2007; M Ramon, T Busigny, G Gosselin and B Rossion, unpublished data). Consequently, it is not clear whether discrimination of changes of the overall facial configuration, as well as the observation of a face geometry effect require extensive (experimental, or real-life) exposure to the face stimuli used. Moreover, it remains to be determined whether the observation of a face geometry effect is confined to situations where stimuli are presented simultaneously.

The present study reports two experiments conducted to address the above questions, testing subjects of an independent, larger cohort than previously (Ramon & Rossion, 2007; M Ramon, T Busigny, G Gosselin and B Rossion, unpublished data). Here, using the same manipulations, the first experiment served to replicate the finding that healthy observers can discern manipulations of the overall facial geometry, and show and advantage for more as compared to less distorting ones (i.e., a face geometry effect) when comparing face stimuli to internally stored facial representations. Observers who were personally familiar with the individuals depicted performed veridicality decisions between simultaneously presented original faces paired with altered versions that involved more or less distorting changes of the overall facial configuration. In line with previous findings (Barton, Zhao & Keenan, 2003; Ramon & Rossion, 2007; M Ramon, T Busigny, G Gosselin and B Rossion, unpublished data), we observed a face geometry effect, i.e., an advantage for discriminating between original and more distorting changes of the facial configuration.

The second experiment sought to determine whether the subjects’ pre-experimentally acquired familiarity with the identities would affect holistic processing—in terms of their discrimination performance in general, and potentially the presence of a face geometry effect. To this end, familiar and unfamiliar observers discriminated the same stimulus manipulations as in Experiment 1 in the context of a delayed matching task. This was chosen deliberately, as simultaneous stimulus presentation of unfamiliar faces for long durations encourage an image, or feature-by-feature matching strategy—the antidote of holistic processing, which operates in particular given short exposure durations (e.g., Davidoff, 1986; Hole, 1994; Stollhoff et al., 2010). Here only familiar subjects were able to perceive differences in changes of the overall facial configuration (as evidenced by their above chance level performance, which controls did not exceed), but did not exhibit a face geometry effect. Together, these results indicate that personal familiarity is associated with more efficient integration of the overall facial configuration, i.e., holistic processing, but that the observation of a face geometry effect may depend on stimuli being presented simultaneously.

Materials and Methods

The personally familiar faces from which stimuli were created were comparably familiar to all individuals of the peer groups tested (senior year psychology master students from the University of Louvain). This procedure has been applied elsewhere (M Ramon, T Busigny, G Gosselin and B Rossion, unpublished data) and bears the advantage of having stimulus sets of personally familiar faces, that are identical across subjects, and larger in size (here: 26 and 14 identities in Experiments 1 and 2, respectively) than those utilized in studies using subject-specific sets of personally familiar faces (e.g., two (Arsalidou et al., 2010) to six (Gobbini et al., 2013) personally familiar identities). Control subjects tested in Experiment 2 were psychology students from a different cohort who were unfamiliar with the identities depicted; all subjects received financial compensation for participation. The experiments were undertaken with the understanding and written consent of each subject, and conform to The Code of Ethics of the World Medical Association (Declaration of Helsinki). Consent for publication was obtained for individuals depicted in the figures exemplifying stimuli used.

Despite offering more control of the stimulus material (all images were taken under identical conditions by the experimenter, as opposed to being provided by subjects; e.g., Gobbini et al., 2013), and enabling larger numbers of trials (with comparably less image repetitions), the approach of using faces derived from the same cohort comes at the expense of a limited number of available potential participants. As previously (Ramon, Caharel & Rossion, 2011; Ramon, in press), classical parametric statistical methods were complemented with robust techniques (percentile bootstrap analyses). Given their higher statistical power and robustness to deviations from the assumed optimal distribution parameters, the latter are particularly recommended, given small sample sizes and/or unknown theoretical distributions of a statistic of interest (Adèr, Mellenbergh & Hand, 2008). To investigate performance differences related to the experimental manipulations, for each behavioral measure subjects were sampled with replacement and differences between the bootstrap populations for the conditions in question were computed. This process was repeated 999 times, leading to a distribution of bootstrapped estimates of the mean differences across conditions. Differences between the sample means were considered significant if the 95% bootstrap confidence intervals (btCIs) did not include zero. Relying on an estimation of the H1, this bootstrap technique tends to have more power than other robust methods that evaluate the null hypothesis (Wilcox, 2012). Across experiments, these analyses were conducted on accuracy scores (proportion correct) and correct reaction times (RTs); effect sizes are provided where differences reached significance.

Experiment 1: Veridicality decisions for personally familiar faces across manipulations of overall facial geometry

Participants. Thirteen subjects who were personally familiar with the faces presented as stimuli (mean age: 24 ± 1; eight female; one left-handed male) participated in Experiment 2.

Stimuli. Full frontal color photographs of 26 students were cropped of hair and external features using Adobe Photoshop. The resulting stimuli (originals; 154–183 pixels wide, 218–256 pixels high) were modified to create four altered face stimuli per identity. The inter-ocular and nose-mouth distances could be either increased or decreased, with changes always applied in conjunction; that is, the eyes were moved either out (EO) or in (EI), and the mouth was moved up (MU) or down (MD), giving rise to one image per experimental condition (EO_MU, EI_MD, EI_MU, EO_MD) per identity. Given the size of the stimuli, the features were displaced by two pixels per feature. The stimulus modifications applied to each original face are illustrated in Fig. 1.

Apparatus and procedure. Stimuli were displayed using Eprime software, on a 19″ monitor (58 cm viewing distance, 1,280 × 1,024 pixel resolution). Stimulus (i.e., face) size comprised 4.5–5.2° (width) and 6.5–7.5° (height) of visual angle (VA), respectively. Participants performed a veridicality decision task: they decided which of two juxtaposed stimuli (presented until response provided) displayed the original face of the identity depicted on a given trial by pressing a left or right key, respectively. The experiment consisted of four blocks of equal length with interleaved pauses. In total, participants completed 208 trials (26 identities × four change types × two possible locations for original faces) which were separated by a 1,000 ms inter-trial interval and randomly assigned to the four blocks (trial presentation randomized). Prior to the actual experiment, participants completed four practice trials (excluded from analysis).

Experiment 2: Delayed matching across manipulations of overall facial geometry for personally familiar and unfamiliar faces

Participants. Two groups of twelve subjects each were tested in Experiment 2. Participants of the experimental group (mean age: 23 ± 1, eight females) and control group (mean age: 20 ± 1, ten females) differed only with respect to their personal familiarity with the identities presented as stimuli (familiar, unfamiliar).

Stimuli. A subset of stimuli created for Experiment 1 was used in Experiment 2; this subset included original (i.e., veridical) faces of 14 individuals and their respective modified versions (see Fig. 1). This smaller stimulus set was chosen given the different procedure and number of trials necessary (see below). Since stimuli were presented individually here and thus could be presented at a larger size (between 185 and 218 pixels in width, and 260 and 307 pixels in height due to inter-individual differences in face size and shape), feature displacements amounted to ∼5 pixels per feature for each of the four possible modified versions of a face.

Apparatus and procedure. Stimuli were displayed using Eprime software, on a 19″ monitor (58 cm viewing distance, 1,280 × 1,024 pixel resolution). Probe and test faces’ height comprised 5.4–6.3° (width) and 7.8–9.0° (height) of VA, respectively. Participants performed a delayed matching to sample task, requiring same/different decisions between the sample and subsequently presented probe stimulus on a given trial by pressing a left or right key, respectively.

Prior to the experiment, participants completed two practice blocks (32 trials each, all excluded from the analyses) throughout which two female identities and their respective altered versions were presented. Subjects were prompted to be as accurate as possible and attempt to achieve over 80% correct. The two blocks differed in that for the first, feedback was provided on each individual trial, while during the second block feedback was provided after the first and second half of the practice block. After both practice blocks, the total score was provided and participants were again encouraged to perform as accurately as possible and maintain high scores.

Throughout the actual experiment, stimuli derived from 12 other individuals faces (six female) were presented. The presentation parameters were identical to those used in practice blocks. A fixation cross was presented for 400 ms, followed by a 200 ms blank and a sample stimulus displayed for 500 ms. After a 600 ms ISI, a probe stimulus was presented for a maximum of 3,000 ms during which responses were recorded. Consecutive trials were spaced by a 900 ms inter-trial interval; the positions of sample and probes were jittered randomly to avoid participants adopting local matching strategies.

Trials required to investigate the research question posed in Experiment 2 were those for which sample and probe stimuli depicted the original face paired with each altered version (EO_MU, EI_MU, EO_MD, EI_MD). In total there were 96 of these (original/altered) trials of interest: 12 identities × four pairs × two orders (original probe followed by altered sample, or vice versa). An additional 96 trials, also requiring a “different” response, were included which involved presentation of two altered versions as samples and probes (altered/altered). These trials were included to avoid subjects performing their decisions by focusing on a single facial region. Altered/altered trials were chosen so that each of the six possible combinations occurred at least once per identity; an additional two trials per identity was chosen pseudo-randomly ensuring that overall all combinations occurred with equal frequency. Note that performance on these catch trials could not be distinguished in terms of “more” or “less” distorting changes, as the consecutively presented stimuli sometimes involved a single change (eyes or mouth, e.g., if EO_MD and EO_MU were presented in succession) or combinations of changes (both eyes and mouth, e.g., if EO_MD and EI_MU were presented consecutively).1 A further 192 trials were included which involved stimulus repetition, i.e., required “same” responses. For each of the 12 identities these “same” trials involved a minimum of three repetitions for each of the five possible stimulus types (i.e., original, EO_MU, EI_MU, EO_MD, EI_MD); an additional trial per identity was chosen pseudo-randomly ensuring that overall each stimulus type was repeated with equal frequency. Note that the catch (altered/altered) and “same” trials were included in order to ensure equal amounts of trials requiring same and different responses, and identify potentially insufficiently high performance for the trials of interest.

Each participant completed 384 trials (excluding practice trials; identical sample-probe combinations used across subjects), with interleaved pauses every 64 (randomly presented) trials, after which feedback on performance was provided. The presence of a face geometry effect was investigated based on, and provided sufficiently high discrimination performance for, the 96 trials of interest involving consecutive presentation of original and altered facial configurations.

Results

Experiment 1: Veridicality decisions for personally familiar faces across manipulations of overall facial geometry

Experiment 1 assessed whether specific alterations of the overall facial configuration would be discriminated more readily than others in personally familiar faces. Note that other authors have used the term ‘facial configuration’ to refer to single distances between individual features (Maurer, Le Grand & Mondloch, 2002), or include other facial information (e.g., face contour; Sergent, 1984; Young, Hellawell & Hay, 1987). Here ‘facial configuration’ refers to the spatial relationship between the most diagnostic internal facial features—the eyes/eyebrows and mouth (see e.g., Haig, 1985; Gosselin & Schyns, 2001; Sadr, Jarudi & Sinha, 2003)—using stimulus manipulations originally introduced by Barton, Zhao & Keenan (2003). Participants who were personally familiar with the individuals depicted performed identity veridicality decisions for image pairs comprising the original face, along with one that was modified with respect to its facial configuration (see Fig. 1). In line with previous findings (Barton, Zhao & Keenan, 2003; Ramon & Rossion, 2007; M Ramon, T Busigny, G Gosselin and B Rossion, unpublished data), modifications that alter the original facial configuration (i.e., are more distorting) were hypothesized to be more readily perceived than those that preserve it (i.e., are less distorting).

Figure 2A displays the group’s mean performance for more and less distorting manipulations of the overall facial geometry. The planned comparison yielded that more distorting changes were more readily rejected than less distorting ones (accuracy: t(12) = 1.49, p = .16; RTs: t(12) = − 3.04, p = .01, r2 = .44). Percentile bootstrapped estimates of the mean difference between distortion types (i.e., more and less distorting alterations of the overall facial geometry) were computed to validate the distortion-related differences in performance observed with canonical statistics. This analysis confirmed the pattern of superior performance for trials on which original faces were presented together with more, as opposed to less distorting versions (btCIs for all measures: accuracy: [−.002, .05]; RTs: [−474, −121]). While the non-significant trends observed for accuracy scores as tested parametrically could be related to the distribution of the data, the results provided by robust statistics confirm the absence of an effect of condition for this behavioral measure (see also Figs. 2B and 2C for differences in number of errors and RT across conditions for individual subjects). Thus, canonical and robust statistics confirmed the hypothesis that more distorting manipulations were discriminated more readily from original faces than less distorting ones.

Figure 2 Results for Experiment 1: veridicality decision task.

(A) Number of errors and normalized RTs (calculated by dividing the mean RT per condition by the mean across conditions) for more and less distorting conditions (bars represent standard errors). Single subjects’ ranked differences in (B) number of errors and (C) RTs between conditions.

Figure 3 Results for Experiment 2.

(A) Mean performance per group for each of the three trial types presented: trials of interest considered for analyses (original paired with altered facial configurations), and catch trials (stimulus repetition or stimuli depicting different alterations of the facial configuration; see Methods). Percentage correct (as opposed to number of errors) is displayed because the total number of trials involving stimulus repetition was equal to the sum of those requiring “different” responses (original/altered, altered/altered). Note that the control subjects’ performance for trials of interest did not differ from chance level. (B) Familiar and unfamiliar subjects’ discrimination performance for trials of interest, separately for more and less distorting changes of the overall facial configuration (bars represent standard errors), which were not associated with significant differences in performance. Single subjects’ ranked differences in (C) number of errors and (D) RTs between conditions.

Experiment 2: Delayed matching across manipulations of overall facial geometry for personally familiar and unfamiliar faces

Experiment 1 demonstrated an advantage for detecting more over less distorting alterations of the overall facial geometry, i.e., a face geometry effect, when subjects performed veridicality decisions between original and altered faces of personally familiar individuals. Following this observation, in Experiment 2 two groups of observers performed a delayed matching task, which did not require face identification but merely same/different decisions between consecutively presented stimuli. Performance for discriminating changes in the overall facial geometry was considered an index of holistic processing, the hypothesis being that personal familiarity with faces leads to increased integration of facial information from across the entire face, manifesting in terms of more efficient perception of global facial geometry, and potentially a face geometry effect.

Although only trials involving original/altered faces were relevant to investigate the emergence of a face geometry effect, subjects’ accuracy for catch trials was inspected to ensure sufficient performance. Figure 3A displays the groups’ mean performance for trials of interest and both types of catch trials. Independent sample t-Tests revealed no differences in the groups’ performance for catch trials involving stimulus repetition, t(22) = .94, ns, or presentation of two altered versions, t(22) = .86, ns. However, for the trials of interest, i.e., stimulus pairs involving original and altered facial configurations, familiar subjects performed significantly better than unfamiliar controls, t(22) = 3.50, p = .001. A X2 test of proportions revealed that the controls’ performance did not differ significantly from chance (p = .24), contrary to that of familiar subjects’ (p < .02). Although only familiar subjects’ data were therefore further analyzed for potential face geometry effects, unfamiliar subjects’ performance for more, as compared to less distorting trials is displayed in Fig. 3B.

The planned comparison yielded no significant difference between discriminating more or less distorting changes (accuracy: t(11) = − .77; RTs: t(11) = − 1.72; both ns). Percentile bootstrapped estimates of the mean difference between distortion types (i.e., more and less distorting alterations of overall facial geometry) confirmed the parametrically observed lack of a difference between less and more distorting conditions (btCIs for all measures: accuracy: [−.07, .02]; RTs: [−88, 3]). Individual subjects’ differences in number of errors and RT across conditions are displayed in Figs. 3C and 3D.

General Discussion

The aim of the present paper was to determine whether the presence of a facial representation in memory, as for personally familiar faces, would lead to differences in holistic processing, which is considered a hallmark of face processing. Contrary to the commonly used paradigms involving combinations of facial information from different identities (Young, Hellawell & Hay, 1987; Tanaka & Farah, 1993), or displacements of individual inter-feature distances (Goffaux & Rossion, 2007; Sekunova & Barton, 2008; Ramon & Rossion, 2010), here a novel paradigm involving manipulations of the overall facial geometry (Barton, Zhao & Keenan, 2003) was applied. The experiments reported were rooted in the idea that repeated real life exposure with personally familiar faces would give rise to differential processing of global facial geometry.

Barton et al.’s (2003) seminal experiment involved an oddity paradigm, in which observers were required to indicate which of three simultaneously presented unfamiliar face stimuli was different from the remaining (identical) two. In their experiment various manipulations of facial information, including changes of feature color, or single inter-feature distances, were discriminated. Additionally, the overall facial configuration was also manipulated via combinations of altered inter-feature distances between the most diagnostic facial features: the eyes and mouth (Haig, 1985; Gosselin & Schyns, 2001; Sadr, Jarudi & Sinha, 2003). Beyond finding more efficient discrimination for the latter, the authors reported that specific changes to the overall facial configuration were more readily discerned: those that distorted, as opposed to maintained the original feature configuration. Importantly, this was observed only in healthy controls, but not in a case of acquired prosopagnosia.

Experiment 1 involved simultaneous veridicality decisions between the original face of personally familiar individuals and a modified version, in which the overall configuration had been altered. Replicating our previous findings under identical experimental conditions (Ramon & Rossion, 2007; M Ramon, T Busigny, G Gosselin and B Rossion, unpublished data), subjects showed a face geometry effect for correct RTs—i.e., superiority for discriminating more, as compared to less distorted versions of the original inter-feature ratios (Fig. 2). The lack of a significant effect for accuracy scores, as reported by Barton, Zhao & Keenan (2003), may be related to the difference in paradigms and tasks applied. In Barton et al.’s (2003) study, subjects had to identify the ‘odd’ one out of a set of three face stimuli. Here, however, subjects had to identify the veridical face of one of their classmates that was presented together with its more or less distorting version. Comparable accuracy scores across conditions were achieved at the expense of prolonged RTs when trials involved less distorting changes of the overall facial configuration; that is, faces presented on these trials required more visual inspection to be correctly rejected as foils deviating from the facial representation stored in memory. As all features were displaced independently, perception of only the eye or mouth location in isolation would not have led to this observed benefit for more, over less distorting changes of the facial configuration. The observation of this face geometry effect is therefore interpreted in terms of subjects perceiving both sources of information simultaneously, i.e., integrating information from across the entire face.

Experiment 2 sought to determine whether personal familiarity would be associated with an advantage for processing the overall facial geometry in the context of a delayed matching task, which could be completed by unfamiliar subjects as well. Theoretically, performance on this task did not require a facial representation stored in memory, and could be achieved based on matching of the sequentially presented visual input. First we assessed whether familiar and unfamiliar subjects could generally discriminate changes of the overall facial configuration (i.e., irrespective of the type of change). Provided sufficiently high performance, their behavior was then analyzed to determine whether they exhibited a face geometry effect as found in Experiment 1. Two interesting observations were made in Experiment 2. First, only familiar observers were able to detect the presence of a difference in face pairs consisting of original and altered versions, while unfamiliar observers performed at chance (Fig. 3A). As both the inter-ocular and nose/mouth distances had to be considered simultaneously, controls’ insufficient performance indicates that they were not capable of doing so. Secondly, no face geometry effect was observed in familiar observers (Fig. 3B). That is, under sequential matching conditions, the presence of a facial representation stored in memory led to enhanced face discrimination performance, but not the observation of a face geometry effect.

At first sight the results of Experiment 2 may seem to contradict those reported by Barton, Zhao & Keenan (2003): contrary to familiar subjects, our unfamiliar healthy observers were not able to discern manipulations of the overall facial geometry. However, two related aspects might account for these seemingly conflicting findings. First, despite using a comparable number of trials, Barton et al.’s (2003) stimuli depicted only two identities, leading to a larger degree of experimental familiarization with their unfamiliar face stimuli which may have facilitated discrimination performance. Moreover, the unfamiliar faces used here were “subjected to judgments of sameness and difference” (Sergent, 1984). This was done in the context of a delayed matching task involving sequential, as opposed to simultaneous stimulus presentation. The introduction of a delay interval between the presentation of probe and sample faces has been associated with a decay of the inner features (e.g., Rock, Halper & Clayton, 1972; Walker-Smith, 1978). This delay is likely to have increased task difficulty for discrimination of subtle changes of the facial configuration applied to the same identity.

Thus, the differences in presentation regime (simultaneous vs. sequential), as well as the extent to which identities were presented repeatedly (higher given fewer identities depicted), may account for the controls’ inability to discern changes of the overall facial configuration. Familiar subjects’ ability to detect these changes on the other hand indicates that the presence of a facial representation in memory minimizes the aforementioned information decay related to the inter-stimulus delay. The results of Experiment 2 therefore suggest that familiar observers, which exhibit a face geometry effect given simultaneous presentation (Experiment 1; Ramon & Rossion, 2007; M Ramon, T Busigny, G Gosselin and B Rossion, unpublished data), can extract information concerning the overall facial geometry given short presentation durations and maintain it more efficiently than unfamiliar subjects, who lack facial representations stored in memory. As stated above, personal familiarity was expected to lead “to increased integration of facial information from across the entire face, manifesting in terms of more efficient perception of global facial geometry.” Since discrimination of these changes—even in the absence of a face geometry effect—relies on observers’ ability to “integrate local spatial information into overall facial structure” (Barton, Zhao & Keenan, 2003), personally familiar subjects exhibited greater holistic processing, i.e., an increased perceptual field of view (see also Rossion, 2008; Rossion, 2009; Van Belle, Lefèvre & Rossion, 2015) than unfamiliar controls.

An open question the present study cannot answer concerns which aspects of real-life familiarity lead to this increased processing efficiency. Previous research suggests that processing of personally familiar faces calls upon different processes than those involved in recognition of famous or experimentally familiarized faces (Tong & Nakayama, 1999; Knappmeyer, Thornton & Bülthoff, 2003; Carbon, 2008). One main difference between personally familiar and unfamiliar, or experimentally learned faces, concerns the degree of variability in the visual input at encoding or learning phases. Experimentally acquired familiarity typically involves image learning (e.g., Caldara et al., 2005; Tanaka et al., 2006; Herzmann & Sommer, 2007; Herzmann & Sommer, 2010; Barsics & Brédart, 2012), and/or a restricted number of different images per identity (O’Donnell & Bruce, 2001; Gobbini & Haxby, 2006; Tanaka et al., 2006), most commonly presented with constant stimulus size. Personally familiar faces on the other hand are encountered across a range of viewing conditions, rendering their representations and therefore identification robust to variations such as changes in viewpoint and exposure across different viewing distances. With varied viewing distance, the visual information projected to the retina changes, leading to differences in the available spatial frequency content (Sowden & Schyns, 2006). As demonstrated in Fig. 4, personally familiar faces can be reliably identified despite large variations of available information associated with changes in viewing distances, and are processed with higher efficiency than their unfamiliar counterparts (Kemp, Towell & Pike, 1997; Bruce et al., 1999).

Figure 4 Personally familiar face identification across simulated viewing distances.

Mean performance and 95% confidence intervals for identification of colleagues’ faces (N = 11; mean age: 32 ± 7; seven female) plotted as a function of viewing distance. The reduced-size images displayed in the bottom row depict stimuli used in the experiment to simulate viewing distances; the top row displays the visual information projected to the retina. Subjects verbally identified gray scaled, full frontal, naturalistic images of 39 members of the Department of Psychology, University of Glasgow, which were taken under identical viewing conditions and matched for luminosity. Faces were centered on a 1,024 × 1,024 pixel canvas with grey background; increasing physical distance between participants and faces to be identified was simulated by shrinking image size with the Laplacian pyramid (Burt & Adelson, 1983). This recursively removes the highest SFs of an image while down-sampling the residual image by a corresponding amount as done e.g., by Smith & Schyns (2009). The simulated viewing distances ranged from 3.3 to 844.8 m, with stimuli recursively down-sampled from an initial on-screen size of 24 cm in height displayed at a 3.3 m viewing distance (see Loftus & Harley, 2004).

These visual input variations during face learning in real-life scenarios may form the basis for familiarity-dependent differential processing manifesting in terms of prioritized detection or increased behavioral recognition speed (Herzmann et al., 2004; Ramon, Caharel & Rossion, 2011; Gobbini et al., 2013). They may also account for familiarity-related differences in perceptual processing suggested by studies of oculo-motor patterns (Van Belle et al., 2010), discrimination of feature displacements (Brooks & Kemp, 2007; Ramon, in press), and spatial frequency thresholds (Watier & Collin, 2009). Importantly, changes in viewing distances lead to perception of an individual’s facial configuration across a range of spatial frequencies, which may be responsible for the increased discrimination sensitivity observed here, and familiarity-dependent differences in processing of vertical inter-feature distances (Ramon, in press).

Further studies are required to address this assumption, ideally involving larger sample sizes to detect potentially small effects. Moreover, a longitudinal approach would be favorable in order to e.g., track the development of participants’ sensitivity to the overall facial configuration throughout the course of familiarization. A further necessary aspect such studies should consider is the systematic control of visual input variations, e.g., viewing distances (Sowden & Schyns, 2006), facial motion (Knappmeyer, Thornton & Bülthoff, 2003) and viewpoint changes (Stevenage & Osborne, 2006) throughout the course of familiarity acquisition.

The present results expand on previous neuropsychological findings that demonstrate the importance of holistic processing for perception of overall facial geometry (Barton, Zhao & Keenan, 2003; Ramon & Rossion, 2007; M Ramon, T Busigny, G Gosselin and B Rossion, unpublished data). They are interpreted as evidence that personal familiarity with faces is associated with increased sensitivity to the overall facial configuration, i.e., holistic processing. Taken together, the results reported here indicate that holistic processing is facilitated by the presence of a facial representation stored in memory.

Supplemental Information

Supplemental Information 1 Raw data

Raw data for Experiments 1 and 2.

Click here for additional data file.

The author expresses her gratitude to Bruno Rossion, under the supervision of which the experiments were carried out. Further thanks are directed to Philippe Schyns and Luca Vizioli, to all participants for their cooperation, as well as two anonymous reviewers for their helpful comments on an earlier version of this paper.

Additional Information and Declarations

Competing Interests

Author Contributions

Human Ethics

1 Investigating the effect of single inter-feature relational changes as a function of region, or as compared to combined changes on discrimination performance is an interesting research question in itself (see e.g., Barton, Zhao & Keenan, 2003; Malcolm, Leung & Barton, 2004). However, this was not the research question addressed here; each of these conditions was represented by fewer trials, as compared to the trials of interest (original/altered), therefore making direct comparisons impossible.

The author declares there are no competing interests.

Meike Ramon conceived and designed the experiments, performed the experiments, analyzed the data, contributed reagents/materials/analysis tools, wrote the paper, prepared figures and/or tables, reviewed drafts of the paper.

The following information was supplied relating to ethical approvals (i.e., approving body and any reference numbers):

The experiments were undertaken with the understanding and written consent of each subject, and conform to The Code of Ethics of the World Medical Association (Declaration of Helsinki). Consent for publication was obtained for individuals depicted in the figures exemplifying stimuli used.

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
