# Peer review of "Perception of global facial geometry is modulated through experience"

_PeerJ, doi:10.7717/peerj.850_

## Round 0.1 · original submission · Major Revisions

Please notice that extensive revision and resizing of aims are required to match the paper's purposes with the experimental design. Please make sure to respond to all reviewers' issues.

Reviewer 1 ·

Basic reporting

The reported procedure is comprehensible and the addressed question is of scientific relevance.

The following abbreviations could be introduced more clearly: Reaction Time (RT), inverse efficiency index (IE).

In the introduction on page 6, line 6, you mention “(here: 26, 14 identities)”. Please add a sentence or so explaining this in more detail. Does it mean 26 identities in experiment 1 and 14 in experiment 2?

Figures 2b & 3b have two ordinates (y-axis) – it is not clear from the display which data refers to which axis. It is also unsual that some data is represented in bars and the other not. Please unify and consider using different ordinate colors for the different types of data.

Style issues:
On page 11, line 3 and 4, the experimental group is specified with twelve participants (eight females) and the control group has twelve participants (two males). I suggest to use a uniform specification i.e. eight females and ten females, respectively.

On page 16, line 21, I do not completely understand the sentence: “our unfamiliar healthy observers did not present with a face geometry effect.” Does this mean: “.. did not show a face geometry effect.”?

Orthography issues:
Page 4, line 17, typo? Consider: PS on the other hand
Page 15, line 3, typo? Did you mean: differential processing of global facial geometry

Experimental design

On page 12, line 16 & 17 the author states: “Per identity there were 16 “same” trials (minimum of 3 repetitions per each of the five possible stimulus types, i.e. original, EO_MU, EI_MU, EO_MD, EI_MD), as well as 16 “different” trials”.
Why did you repeat most of the stim types three times, but one four times? How is the additional stimulus distributed over subjects? Is it always the same type?
How were the 16 “different” trials selected?
There are 10 combinations of “different” faces of one identity:

original - EO_MU - More distorting
original - EI_MU - Less distorting
original - EO_MD - Less distorting
original - EI_MD - More distorting
EO_MU - EI_MU
EO_MU - EO_MD
EO_MU - EI_MD
EI_MU - EO_MD
EI_MU - EI_MD
EO_MD - EI_MD

It is clear that all 4 possible original/altered types are used two times (in two possible sequences: original/altered and altered/original). Are the residual 8 stimulus type combinations chosen randomly?

Validity of the findings

Please provide RTs with the corresponding units. Only fig 3b mentions milliseconds.
The two positions after decimal point in the confidence intervals of RTs and IEs are not relevant, please round this.
Please provide the formula for the IE in addition to the reference.
Is the unit of your IE also milliseconds or do you use a normalization?
Please provide the units in general and consider using percent if you refer to a relative measure.
Furthermore, the term accuracy is not clearly defined, a simple formula or statement like hit rate, true positive rate or true negative rate would clarify this.

Page 9, line 15 & 18: the p-values for accuracy dot not show a significant difference under usual statistical significance thresholds. Is this due to a non-gaussian distribution of this parameter? Please refer to this as an observed tendency in accuracy or similar.

Page 9, line18 & page 14, line 7: which percentiles were used?
Page 9, line 18 & page 14, line 7: t(1, 12) & t(1, 11) is usually written as t(12) & t(11).

Page 14, line 10: the confidence interval for the differences of RTs seems to contradict with the above stated observed lack of a difference. There is an obvious asymmetry around zero of about 3/90. What is the probability that the difference is less than zero? This would be interesting for the difference of accuracy as well.

Additional comments

For the general understanding of the statistics, figures of histograms for the RTs and accuracies or the IE would help. Figure 1 presents a result from a different study with different stimuli alterations and could therefore be simply removed from the reported study. A reference to it would suffice.

Reviewer 2 ·

Basic reporting

No Comments

Experimental design

There are interesting ideas here, but it’s a little bit unclear to me what the fundamental research questions are. The hypothesis given in the abstract is that facial reps stored in memory are associated with differential perceptual processing of overall facial geometry. The intro takes some time getting to outlining this face geometry effect, instead first focussing on differences between familiar and unfamiliar faces, and differences between personally familiar and otherwise familiar faces. These latter differences are not explicitly manipulated in the experiment and serve only as a putative reason why differences in performance between personally familiar and otherwise familiar faces might arise. Thus, for example, the focus on how global facial geometry is maintained over different viewing distances is fine as an explanation of why it is a useful property, but viewing distances are not manipulated in the present experiment and their contribution to the pattern of results is unclear.

A better structure to me would seem to be to outline some of the more general points (i.e. holistic processing and face geometry) and then explain why they might differ between personally familiar and unfamiliar faces. That then leads you to the research question, which would seem to be “does the face geometry effect differ between personally familiar and unfamiliar faces?” The author already reviews evidence which says there are similar effects for both familiar and unfamiliar faces (pg 3-4), in which case we need a clearer rationale for running the current experiments – if that rationale is to replicate these findings and/or test them together, that’s perfectly fine – but it needs to be made clear.

If we run with this research question, then the first experiment does not test it, because all faces are personally familiar; so how does it fit in the picture? I see that the author explains that Exp 1 tests if larger modifications of overall facial configuration are more easily discriminable from the original than smaller modifications. So the first experiment really serves to establish that you can find similar effects as before with this stimulus set and this group of people. Again, this is fine, but the author needs to explain how this fits in the story.

Now, it’s only the second experiment that directly tests the main research question. Having two separate groups of participants – one for whom the the faces are personally familiar, one for whom the faces are not familiar – neatly sidesteps the argument that low-level differences in the faces underpin differences in performance, rather than familiarity per se, since the same faces are presented to each group.

However, Exp 2 also introduces a *new* question: does the facial geometry effect depend on the task? This is a good question in itself, and Exp 1 would motivate it well, though the authors should explain why switch to this particular task. But Exp 2 doesn’t really test this new question, because it does not directly contrast performance on different tasks (i.e. simultaneous vs sequential presentation). In any case the unfamiliar participant group is not required to test it.

In fact if the sequential task does not produce the expected facial geometry effect (which it doesn’t appear to from the results – though see later comment), then it cannot be used to answer the main research question “does the face geometry effect differ between personally familiar and unfamiliar faces?”, since it does not produce a face geometry effect for anyone.

12 participants in each group – and I understand that your participant base is necessarily limited – is very low. The difference in performance would have to be very large for this to be a sufficient group size.

Minor

Experiment 1 – how long were the stimuli presented for?

The “general methodological considerations” section can probably be folded into the methods for Experiments 1 and 2 rather than needing to be in their own section.

Validity of the findings

I have a number of concerns about the analyses used throughout. Most importantly, the author uses a one-way ANOVA with four levels to test for overall differences between stimulus conditions, both in Exp 1 and in Exp 2 (for the “familiar” group). However, a 2 x 2 RM ANOVA would seem more appropriate, with factors Eyes (out or in) and Mouth (up or down). Thus one could tell if manipulating the eyes was more important, or manipulating the mouth, or if the two things interacted. The interaction term would answer the question the planned comparisons set out to test – if less manipulated images are harder to discriminate than more manipulated images – as this should show up as a cross-over interaction.

Another alternative would be to simply skip the omnibus ANOVA – it is not strictly necessary when one has a clear a priori comparison to make. However, I’d strongly recommend using the ANOVA framework. Then, for experiment 2, you can simply add group as an additional, between-subjects factor (so you can test 2 x 2 x 2 – group x eye x mouth). This allows you to directly test whether the facial geometry effect differs across groups, which is not actually tested at present. I suspect, however, that 12 subjects will be unlikely to provide the significant three-way interaction you would be hoping for - especially with this task.

Given that the incorrect procedures are used, it’s hard to assess the accuracy of the results. I think the data needs to be reanalyzed before a proper assessment of the either the results or the general discussion can take place. But the author should bear in mind some of the limitations imposed on interpretation by their design choices, as outlined above.

I would not recommend analyzing inverse efficiency ; It does not offer any benefit over analyzing reaction times and errors, see http://www.psychologicabelgica.com/article/view/pb-51-1-5 and http://cogsci.stackexchange.com/questions/660/how-to-analyze-reaction-times-and-accuracy-together/668#668

Minor
Please start bar graphs from zero. I’d also recommend not combining the RT and accuracy plots. I usually plot errors rather than accuracy when also looking at RTs, because then both go in the same direction: lower numbers = better performance. It’s thus easier to compare patterns across the two measures.

Furthermore, perhaps signal detection methods, such as d-prime, would also help account for response biases and so on, rather than analyzing accuracy/error rates per se.

I’m not sure I follow some of the logic in choosing not to analyse “altered-altered” trials “Performance on these trials could not be distinguished in terms of “more” or “less” distorting changes,” – perhaps not, but for example one might predict that a change of two features would be more easily discriminable than a change of a single feature? Perhaps the most easily distinguishable of all would be a change from one of the “most” distorted to the other “most” distorted face?

Additional comments

There are nice ideas present here; but it really needs a more coherent explanation of what is being tested and why, and unfortunately i'm not sure at the moment the experiments here can provide a strong answer to any of the main questions.

---

## Round 0.2 · Major Revisions

I agree with the reviewers that the manuscript underwent substantial changes. However, I agree with Reviewer 2's profound criticisms.

I suggest the authors implement each observation with the greatest attention, give the paper a clear direction and a logic flow, and restrict result interpretation to what data say.

Reviewer 1 ·

Basic reporting

The author implemented the suggestions of the reviewers well in the field of Basic Reporting.

Experimental design

For me, this is clear now.

Validity of the findings

This was greatly improved by the implementation of the suggestions from reviewer 2

Additional comments

I agree with your profound response to the reviewers comments.
Only one minor detail remains unclear to me: Why do you emphasize the Errors in your figures 3a and 4b with bars while the RTs are plotted by a line. I would prefer a uniform plot with either (bars, bars) or (line, line).

Reviewer 2 ·

Basic reporting

No comments

Experimental design

I commend the author for the effort she has made to change the introduction. Unfortunately, I think the new introduction still has problems, some old, some new. I still cannot quite follow the arguments being made here.

Here are some of the specific issues I currently have.

Pg3 lines 5-8. What is a veracity decision?

Pg 3, lines 10-12. I don’t see how this sentence relates to the rest of the paragraph.

Pg3, lines 12-17. The Ramon studies and Barton studies differ on stim material and task. Why does it matter when both found similar results? The way this paragraph is written is as if they produced different results which may be down to these differences in stimuli and task. In fact you’re arguing that the similarity between the results is down to an unacknowledged limitation of the Barton design, namely that they have many trials with the same identities, so people become experimentally familiarized with the faces. You then want to argue that it hasn’t been demonstrated with *truly* unfamiliar faces.

Pg 4-5, 2nd paragraph onwards. You argue that variability in the visual input at encoding is a key factor underlying differences between personally familiar and unfamiliar faces, and then go on to explain how these may underlie familiarity-dependent differential processing of faces. Specifically, changes in viewing distance do not alter the overall configuration of faces, and thus learning stable facial geometry may underpin the ability to cope with variation (due to viewing distance etc).

However, I still can’t see how, in context, this helps motivate the present experiments. There are no differences between the Barton (with unfamiliar faces) and Ramon (familiar faces) studies which need to be explained away. Changes in viewing distance cannot explain why the effect was present in the Barton study, since there were no changes in viewing distance etc., as you already mention on pg4 (lines 6-10). Thus it seems unlikely that the face geometry effect observed there is due to variations in the perceptual input at encoding. And, again, you do not manipulate viewing distance etc.

Ultimately, it seems the argument you want to make is that faces which are unfamiliar and do not become experimentally familiar may not show the face geometry effect. But the rationale behind this still does not seem so clear to me.

Pg 5, lines 11-14. You still do not really explain what experiment 1 is for.
Pg 5, lines 16-18. You still do not explain why change to a delayed matching task. Why might simultaneity of presentation be important?

Finally, my understanding is that you want to test whether the face geometry effect depends on familiarity. You define the effect on pg 2 as being the finding that discrimination performance improves as the severity of distortions increases. You do not find this effect in Experiment 2. Thus, again, you can’t demonstrate with a task which does not produce the face geometry effect, as you defined it on pg2, that the face geometry effect differs between familiar and unfamiliar faces.

Validity of the findings

In the discussion, pg 16, 1st paragraph. The key conclusion for Experiment 1 is that participants showed superior discrimination for more distorted faces than less distorted faces. This isn't significant for accuracy/errors. This should be discussed in the context of previous findings.

Pg 17, lines 5-6. You state that unfamiliar healthy observers did not show a face geometry effect, but, as I said in the original review, you never explicitly test this, and you do not present the data from the unfamiliar participants, so it's impossible for us to evaluate this interpretation. I appreciate that you feel it is unnecessary to do so when they performed at chance on overall discrimination, but perhaps there is a cross-over interaction and no main effect of geometry?

Figures 3 and 4 show, respectively, normalized RTs and un-normalized RTs. Why are they normalized for one figure and not for the other? Second, perhaps I misunderstand how the normalization was done, but according to the description you normalized by dividing mean RT with the sum of RTs across conditions. How can this yield RTs over 1, since a mean RT can never be higher than the sum of RTs?

The discussion of Experiment 2, beginning pg 17, and particularly the new section towards the end of the discussion, is welcome and well-measured.

Ultimately, I think the evidence presented here is, overall, rather weak. In experiment 1, only one of the key contrasts is significant. In experiment 2, very little is significant at all, which may indicate the study has serious problems with low statistical power, given that a lot ends up resting on null effects. I think the author really needs to think about what the focus of the paper needs to be, and be prepared to make fewer firm statements about what can be concluded.

---

## Round 0.3 · accepted · Accept

I am willing to accept the present version of this manuscript. While I stand by some of the objections of reviewer 2, I also see the validity of some of the points in the Author's rebuttal. I am quite certain that a further round of revisions would not change either position. However, as the merits and limits of the paper have become self-evident, and the methods are accurately reported, I deem the work ready for being judged by the reader him/herself.